# Application of a Fractional Order Differential to the Hyperspectral Inversion of Soil Iron Oxide

Hailong Zhao [1], Shu Gan [1,2,*], Xiping Yuan [2,3], Lin Hu [1], Junjie Wang [1] and Shuai Liu [1]

[1] Faculty of Land Resources Engineering, Kunming University of Science and Technology, Kunming 650093, China
[2] Application Engineering Research Center of Spatial Information Surveying and Mapping Technology in Plateau and Mountainous Areas Set by Universities in Yunnan Province, Kunming 650093, China
[3] College of Geosciences and Engineering, West Yunnan University of Applied Sciences, Dali 671000, China
[*] Correspondence: gs@kust.edu.cn

**Abstract:** Iron oxide is the main form of iron present in soils, and its accumulation and migration activities reflect the leaching process and the degree of weathering development of the soil. Therefore, it is important to have information on the iron oxide content of soils. However, due to the overlapping characteristic spectra of iron oxide and organic matter in the visible-near infrared, appropriate spectral transformation methods are important. In this paper, we first used conventional spectral transformation (continuum removal, CR; standard normal variate, SNV; absorbance, log (1/R)), continuous wavelet transform (CWT), and fractional order differential (FOD) transform to process original spectra (OS). Secondly, competitive adaptive reweighted sampling (CARS) was used to extract characteristic wavelengths. Finally, two regression models (backpropagation neural network, BPNN; support vector regression (SVR) were used to predict the content of iron oxide. The results show that the FOD can significantly improve the correlation with iron oxide compared with the CR, SNV, log (1/R) and CWT; the baseline drift and overlapping peaks decrease with increasing the order of FOD; the CARS algorithm based on 50th averaging can select more stable characteristic wavelengths; the FOD achieves better results regardless of the modelling method, and the model based on 0.5-order differential has the best prediction performance ($R^2$ = 0.851, RMSE = 5.497, RPIQ = 3.686).

**Keywords:** soil; hyperspectral; iron oxide; spectra transform; fractional order differential



## 1. Introduction

Iron oxide is the bulk of iron-bearing minerals in soils, mainly formed by the chemical weathering and the redeposition of iron-bearing silicate minerals, and is widely distributed in various types of soils around the world [1]. Due to its high activity, the morphological characteristics of iron oxide are susceptible to various environmental factors, and its aggregation and migration activities reflect the leaching process, the degree of weathering development and the zonation of the soil distribution [2]. The chemical activity of iron oxide allows it to adsorb numerous heavy metals, non-metallic ions and oxygenated anions, which greatly control the concentration, morphology and migration transformation of these elements in the soil, determining plant effectiveness, environmental toxicity, affecting crop yield and quality and human health [3].

The traditional method for the determination of iron oxide content has a high accuracy but a high determination cost and a long cycle time. Hyperspectral, with its high spectral resolution and wavelength continuity [4], is widely used for the inversion of soil physico-chemical properties [5,6]. In practice, however, many factors can affect the quality of the spectra; these include the complexity of the composition of the soil itself, the environment and the noise of the instrument itself. Therefore, suitable spectral pre-processing is an indispensable step in soil hyperspectral modelling to improve the predictive power of the

model [7]. Common spectral pre-processing methods include spectral smoothing (Savitzky–Golay filter, SG) [8], continuum removal (CR) [9], absorbance (log (1/R)), multiple scattering correction (MSC) [10], standard normal variate (SNV) [11], continuous wavelet transform (CWT) [12] and differential transformations [13]. Due to the presence of environmental and instrumental noise, spectral smoothing has become an essential step and other processing methods are based on spectral smoothing afterwards. Of these pre-treatment methods, CR, log (1/R), SNV and CWT have all been widely used. However, as soil spectra are a comprehensive reflection of soil properties, the characteristic wavelengths of iron oxide are not the same in different regions and are easily masked by organic matter. According to previous studies, the characteristic spectra of iron oxide and organic matter often overlap in the visible-near infrared band (400–1000 nm) [14]. Spectral differentials can minimize baseline drift and separate overlapping spectra. Of these, first-order and second-order differentials are two effective methods [15,16]. However, the integer order differential lacks sensitivity to the asymptotic slope or curvature of the spectra, resulting in detailed spectral information not being captured [17]. Fractional order differential (FOD) is an extension of the integer order differential, which allows us to interpolate between the original spectra (OS), the first-order differential spectra and the second-order differential spectra and even higher order differential spectra to obtain fractional order differentials. At present, FOD has been widely used in soil hyperspectral and has achieved good results. Tian et al. [18] collected soils from Xinjiang and determined the total salt content of the soils indoors. Firstly, FOD was used for five transformed spectra, and the bands whose spectra and total salt content passed the 0.01 significance test were extracted as characteristic wavelengths and finally modelled using PLSR. The prediction results showed that the best model prediction ability was obtained based on the model of 1.6-order. Hong et al. [19] collected soil samples and measured the organic matter content in the Jianghan Plain of Wuhan City, Hubei Province, while performing FOD on the original spectra at 0.25-order intervals, and the experimental results showed that the PLS-SVM model constructed based on 1.25-order had the strongest predictive power. However, so far, no studies have been carried out to estimate the iron oxide content using FOD.

Due to the high number of wavelengths in the hyperspectral, the wavelength information contained tends to be more redundant. If the full wavelength band is modeled, it not only increases the running time, but also reduces the accuracy of the model [20]. Therefore, the selection of the characteristic wavelengths before modelling is a very important step. Currently, the selection of characteristic wavelengths is mostly completed using the Pearson correlation analysis [21–23], and correlation coefficients and significance levels reflect the correlation between soil physicochemical properties and wavelength [24]. In addition, the Genetic Algorithm (GA) [25], uninformative variable elimination method (UVE) [26], successive projections algorithm (SPA) [27] and competitive adaptive reweighted sampling (CARS) algorithm [28] are the common methods used for the selection of characteristic wavelengths. The wavelengths obtained using these methods are used as the input variables to the model and the iron oxide content is used as the dependent variable for model construction. There are also many methods of model construction, such as multiple linear regression (MLR), partial least squares regression (PLSR) and principal components regression (PCR), all of which are linear regression methods and are simple to use. Of these, PLSR is the most common regression method. Xiong et al. [16] used PLSR to invert the Fe of soils and achieved a good prediction accuracy. In addition, with the popularity of machine learning, more and more methods such as random forest regression (RFR), support vector regression (SVR) and back propagation neural network (BPNN) have been applied to soil hyperspectral modelling. Qin et al. [29] used RFR to model the inversion of free iron in soil and found that the accuracy of RFR in estimating free iron in soil was significantly better than that of stepwise multiple linear regression.

However, due to the complexity of soils in different regions, there is no universal pre-processing method that is suitable for different regions. Therefore, this paper uses three types of spectral transformation methods, including conventional transform spectra

(CR, log (1/R) and SNV), CWT and FOD to transform the OS. The CARS was used to select characteristic wavelengths. Finally, the model was constructed using BPNN and SVR. Therefore, the objectives of this paper are (a) to explore the model prediction capability of the fractional order differential transformation and to compare it with the conventional spectral transform, the continuous wavelet transform; (b) to assess the capability of CARS in characteristic wavelength selection; and (c) to Compare the predictive power of BPNN and SVR models with different spectral transforms.

## 2. Materials and Methods

### 2.1. Study Area

The study area is located in Lufeng County, Chuxiong Yi Autonomous Prefecture, Yunnan Province ($24°55'25''$~$25°22'05''$ N, $102°00'00''$~$102°9'00''$ E). The study area is about 6 km wide from east to west, 8 km long from north to south, and 6 km in diameter, covering an area of about 32 km$^2$, with an overall depression pit with a high elevation around and a low elevation in the middle. The area is a small Mesozoic red sedimentary basin, belonging to the Lower Ordovician Redstone Shale Formation, with a brief lithology of purplish-red and grey–green siltstone. According to two soil surveys in 1982 and 1985, there are five soil types, ten subtypes, twenty genera and forty species of brown soil, yellow–brown soil, red soil, purple soil and rice soil in Lufeng County. The purple soil accounts for 56.9% of the land area and is the most important soil type in the area, followed by red soil, which accounts for 22.8% of the land area, yellow–brown soil, which accounts for 7.8%, and the rice soil, which accounts for 6.3% [30].

### 2.2. Sample Collection and Data Acquisition

Soil samples were collected at the end of July 2021 from the southern rim of the Dinosaur Valley in Lufeng County, Yi Autonomous Prefecture of Chuxiong, Yunnan Province, and the sampling points were set up according to the difference in topography. Each sample was taken within 5 m × 5 m. Figure 1 shows the location of the sampling points in the study area. Within the sampling area, surface soil was collected from 0 to 20 cm according to the 5-point sampling method, and approximately 1 kg of soil was bagged and stored. The soil types were purple loam, red loam and yellow–brown loam. The collected soil was first cleaned of impurities such as weeds and stones, then naturally air dried and finally ground with an agate ball mill and sieved through 100-mesh. The aperture size of 100-mesh was 0.15 mm. Each sample was split into two, one for the determination of iron oxide content and the other for the measurement of hyperspectral data. The determination of iron oxide in soil was carried out by X-ray fluorescence spectrometry in accordance with the "Methods of Agricultural Chemical Analysis of Soil", taking into account the quality requirements of the samples and other technical specifications, as well as the limits of detection, accuracy and precision of the samples.

Soil spectroscopy was carried out in a dark room with an ASD Field Spec 3 geophysical spectrometer, using a probe with an internal halogen light source, a 21 mm inner probe diameter and a 25° front field of view and a wavelength range of 350–2500 nm. The number of wavelengths obtained by resampling the spectral interval to 1 nm was 2151. For the spectral measurements, the soil samples were placed in a 10 cm wide and 2 cm high container and scraped flat to reduce the effect of the roughness of the soil sample on the spectral measurements. The probe was held at a height of 2 cm from the soil sample and aligned vertically with the sample. Five spectral curves were measured for each sample in the same area. The actual spectral reflectance of the sample was averaged over the five spectral curves.

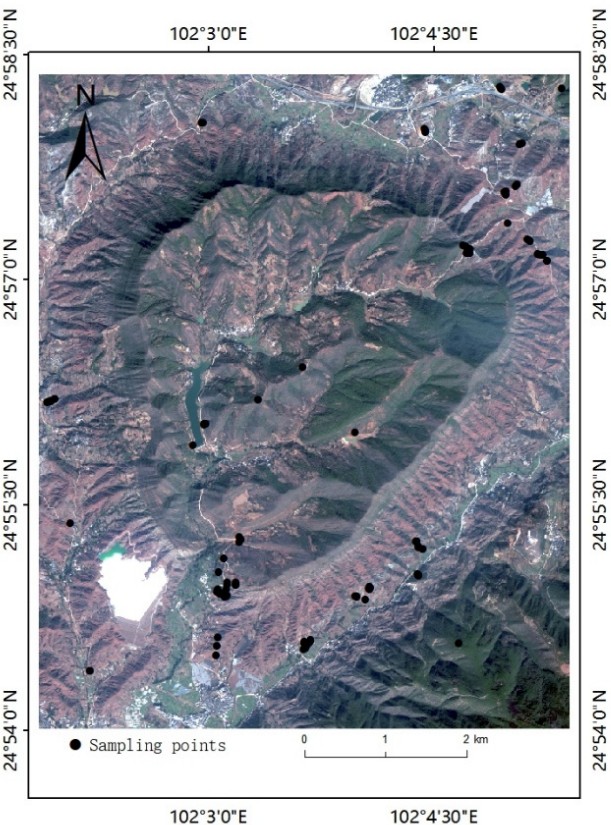

**Figure 1.** Distribution map of the soil sampling points.

### 2.3. Data Processing

The signal-to-noise ratio at 350–399 nm and 2451–2500 nm was low due to the influence of the instrument itself, so these two spectral data were removed and a total of 2051 wavelengths were obtained. To eliminate the interference of instrument noise, uneven distribution of soil particles and random factors, the Savitzky–Golay [8] smoothed curve with a window number of 9 and polynomial order of 2 was used as the OS. The CR, log (1/R), SNV and CWT were further applied to the OS. The CR can highlight the absorption and reflection features of the spectra [31]. The log (1/R) can reduce the interference of multiplicative factors caused by light transformation [32]. The CWT, on the other hand, can mine the characteristic information of the spectra at different scales [33].

Fractional order differential is an extension of integer order differential and is commonly known as Riemann–Liouville, Grünwald–Letnikov and Caputo, of which Grünwald–Letnikov is the most commonly used form.

Before giving the definition of Grünwald–Letnikov, let us observe the formula for the first order derivative:

$$\frac{d^1}{dt^1}f(t) = \lim_{h \to 0} \frac{1}{h}[f(t) - f(t-h)] \tag{1}$$

In Equation (1), the h represents the increment of the spectral variable. From the first-order differential, the second-order differential formula can be derived as follows:

$$\frac{d^2}{dt^2}f(t) = \lim_{h \to 0} \frac{1}{h^2}[f(t) - 2f(t-h) + f(t-2h)] \tag{2}$$

Looping the above method, the nth order differential of the function can be derived as follows:

$$\frac{d^n}{dt^n}f(t) = \lim_{h \to 0} \frac{1}{h^n} \sum_{j=1}^{n} (-1)^j \binom{n}{j} f(t-jh) \tag{3}$$

In Equation (3), the j represents the difference between the upper and lower limits of the derivative. Using the Gamma function to replace the binomial coefficients of Equation (3), while extending the integer order to non-integer order, one can then obtain the $\alpha$-order fractional order differential formula:

$$\frac{d^{\alpha}}{dt^{\alpha}}f(t) = \lim_{h \to 0}\frac{1}{h^{\alpha}}\sum_{j=0}^{[(t-t_0)/h]}\frac{(-1)^{j}\Gamma(\alpha+1)}{j!\Gamma(\alpha-j+1)}f(t-jh) \tag{4}$$

Since the sampling interval of the spectrum is 1, set h to 1. h represents the differential step, t represents the upper limit of the differential, $t_0$ represents the lower limit of the differential. The Gamma function is defined as:

$$\Gamma(z) = \int_0^{\infty}e^{-t}t^{z-1}dt = (z-1)! \tag{5}$$

Then, Equation (4) can be converted to:

$$\frac{d^{\alpha}}{dt^{\alpha}}f(t) \approx f(t) + (\alpha)f(t-1) + \frac{(-\alpha)(-\alpha+1)}{2}f(t-2) + \ldots + \frac{\Gamma(-\alpha+1)}{j!\Gamma(-\alpha+j+1)}f(t-j) \tag{6}$$

In Equation (6), $\alpha$ represents the order. $\alpha = 0$ represents OS; $\alpha = 1$ represents the first-order differential; $\alpha = 2$ represents the second-order differential. The implementation of the fractional order differential in this study was implemented using the FOTF toolbox based on MATLAB 2020b [34].

The workflow for data processing is shown in Figure 2.

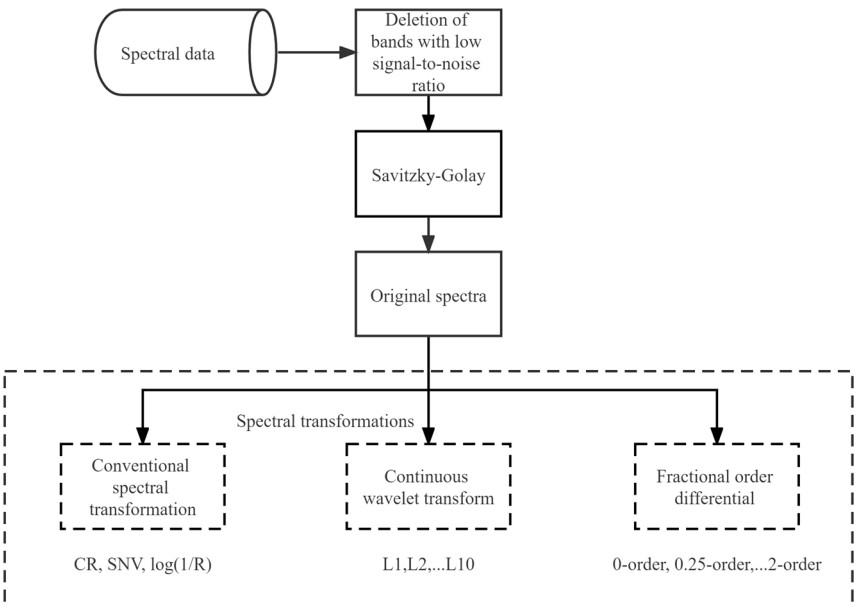

**Figure 2.** The workflow of data processing.

### 2.4. Characteristic Wavelength Selection

Due to the large redundancy of hyperspectral data, not all wavelengths are beneficial in improving the modelling accuracy when performing regression analysis. If all wavelengths are modelled and analyzed, it is not only computationally intensive, but also reduces the modelling accuracy. Therefore, characteristic wavelength selection is necessary prior to modelling.

The Competitive Adaptive Reweighted Sampling (CARS) algorithm is a characteristic wavelength selection method based on Monte Carlo sampling and PLS regression coefficients, treating each variable as an individual and selecting the one with the higher adaptive

capacity. The specific steps are: randomly select a fixed rate of samples as the calibration dataset and build a PLS model, then calculate the absolute value of the regression coefficient of the model and the weight corresponding to each wavelength, use the exponential decay function and adaptive reweighted sampling method to select the variables, while calculating the root mean square error of cross-validation, after sub-sampling, select the subset with the smallest root mean square error as the optimal subset of variables [35].

### 2.5. Support Vector Regression (SVR)

SVR is a non-linear modelling method based on statistical learning theory. Its basic approach is to use the support vectors in the training samples to design an optimal decision boundary to deal with linear and non-linear problems, which performs well especially when dealing with small sample data [36]. In this study, the kernel function of the support vector machine was chosen to be a Gaussian kernel function. To make the model more stable and the results more reliable, the parameters of the support vector machine: c and gamma tuning were performed during the model training process using a genetic algorithm based on differential evolution, which is more robust than the traditional genetic algorithm, has a block convergence speed and has a stronger global optimization search capability. Among the parameters of the differential evolution-based genetic algorithm, the range of values for c and gamma was set between $2^{-8}$ and $2^8$, the population size was 50, the coding method was real number coding, the selection method of the basis vector was elite replication selection, the variation operator F was 0.5, the crossover operator CR = 0.5, the maximum number of evolutionary generations was 1000 and the fitness function was the average root mean square error of cross-validation. The algorithm was implemented using the Geatpy [37].

### 2.6. Back Propagation Neural Network (BPNN)

BPNN is a more widely used artificial neural network, with a strong non-linear processing capability [38]. The main features of BPNN are the forward transmission of input data and the backward propagation of errors. In the forward transfer process, the input data are processed progressively from the input layer through the hidden layer to the output layer. If the error in the output layer is not within the range, back propagation is performed and the weights of each layer are adjusted by gradient descent until the error is within the specified range. In this study, we used a simple three-layer network structure with an input layer, a hidden layer and an output layer, respectively. Transigmoid and purelin were chosen as the transfer functions of the hidden and output layers according to the previous study when building the BPNN prediction model. Besides, Sigmoid and trainlm were chosen as the activation function and training function, respectively [39]. Hidden layer, learning rate and maximum epoch were 8~10, 0.01 and 1000, respectively.

### 2.7. Model Evaluation Method

The Kennard-Stone (K-S) [40] algorithm was used to classify the calibration dataset and the validation dataset. A total of 70% of the samples were selected as the calibration dataset and the remaining 30% as the validation dataset. Since the soil samples showed non-normal distribution, the ratio of performance to interquartile spacing (RPIQ) could give a more realistic evaluation of the model [41]. The accuracy of the inverse model was therefore measured by three parameters: coefficient of determination ($R^2$), root mean square error (RMSE) and RPIQ.

$$R^2 = 1 - \frac{\sum_{i=1}^{n}(y_i - y_i^*)^2}{\sum_{i=1}^{n}(y_i - \overline{y})^2} \tag{7}$$

$$RMSE = \sqrt{\frac{\sum\limits_{i=1}^{n}(y_i - y_i^*)^2}{n}} \tag{8}$$

$$RPIQ = \frac{IQ}{RMSE} \tag{9}$$

where: $y_i^*$ is the predicted of the ith sample; $y_i$ is the measured value of the ith sample; $\bar{y}$ is the mean of the measured values; IQ is the difference between the third quartile and the first quartile of the sample; n is the number of samples. A larger $R^2$ indicates a more stable model; a smaller RMSE indicates a more accurate model; a larger RPIQ indicates a better predictive power of the model [42]. The performance of the models can be judged as follows:

RPIQ: (1) >2.5, excellent model; (2) 2.0–2.5, very good model with predictive ability; (3) 1.7–2.0, good model; (4) 1.4–1.7, fair model; and (5) <1.4, poor model [43].

## 3. Results

### 3.1. Statistical Analysis of Iron Oxide Content

The 135 samples were divided into two groups using the Kennard-Stone algorithm, with 70% being the calibration dataset (*n* = 95) and 30% being the validation dataset (*n* = 45). The obtained soil iron oxide content was counted by origin software and the relevant statistical parameters obtained are shown in Table 1. The minimum value of iron oxide in the study area was 18.293 g·kg$^{-1}$ and the maximum value was 66.978 g·kg$^{-1}$, with a mean value of 41.201 g·kg$^{-1}$ and a coefficient of variation of 28.4%, which is a medium variation. The difference between the mean and standard deviation of the calibration dataset and the validation dataset was not significant, and they can be considered as belonging to the same distribution.

**Table 1.** Statistical characteristics of iron oxide content.

| Sample Classification | Sample Number | Max/(g·kg$^{-1}$) | Min/(g·kg$^{-1}$) | Mean/(g·kg$^{-1}$) | Standard Deviation/(g·kg$^{-1}$) | Coefficient of Variation/% |
|---|---|---|---|---|---|---|
| Total dataset | 135 | 66.978 | 18.293 | 41.201 | 11.698 | 28.393 |
| Calibration dataset | 95 | 64.808 | 23.311 | 42.141 | 10.736 | 25.476 |
| Validation dataset | 40 | 66.978 | 18.293 | 38.969 | 13.605 | 34.912 |

### 3.2. Spectral Transformation Methods

#### 3.2.1. Conventional Transform Spectra

The results of the conventional spectral transformation of the original spectra are shown in Figure 3. After the original spectra were transformed by the continuum removal, the spectral curves were normalized to a consistent spectral background and effectively highlighted the absorption features of the spectra [44], with significant iron oxide absorption features at 500 nm and 900 nm, respectively [45]. Meanwhile, near 1400 nm was the band spectra of the -OH, 1900 nm was the band of $H_2O$, and the absorption feature at 2200 nm was mainly due to the -OH stretching vibration and the AL-OH bending vibration [46,47].

#### 3.2.2. Continuous Wavelet Transform

The Gaussian4 function was selected as the wavelet basis function in this study because the soil spectral curve characteristics were similar to those of the Gaussian function [48]. Based on the calibration dataset, the original spectra were first transformed into corresponding wavelet coefficients (decomposition scales were set to $2^1$, $2^2$, $2^3$, ... , $2^{10}$), and average them. The first scale was denoted as L1, the second scale as L2, and the mth scale as Lm. The results are shown in Figure 4. It can be noticed that the absorption and reflection characteristics increased with increasing the scale in the different wavelength ranges. The wavelet coefficient curves at the L1, L2 and L3 were less distinctive and were approximately straight lines. At L4, L5, L6 and L7, distinct peaks can be observed. At L8,

L9 and L10, convex smooth curves can be observed with a smaller number of peaks. In summary, the CWT can help to highlight features of the spectra and to fully explore subtle spectral features.

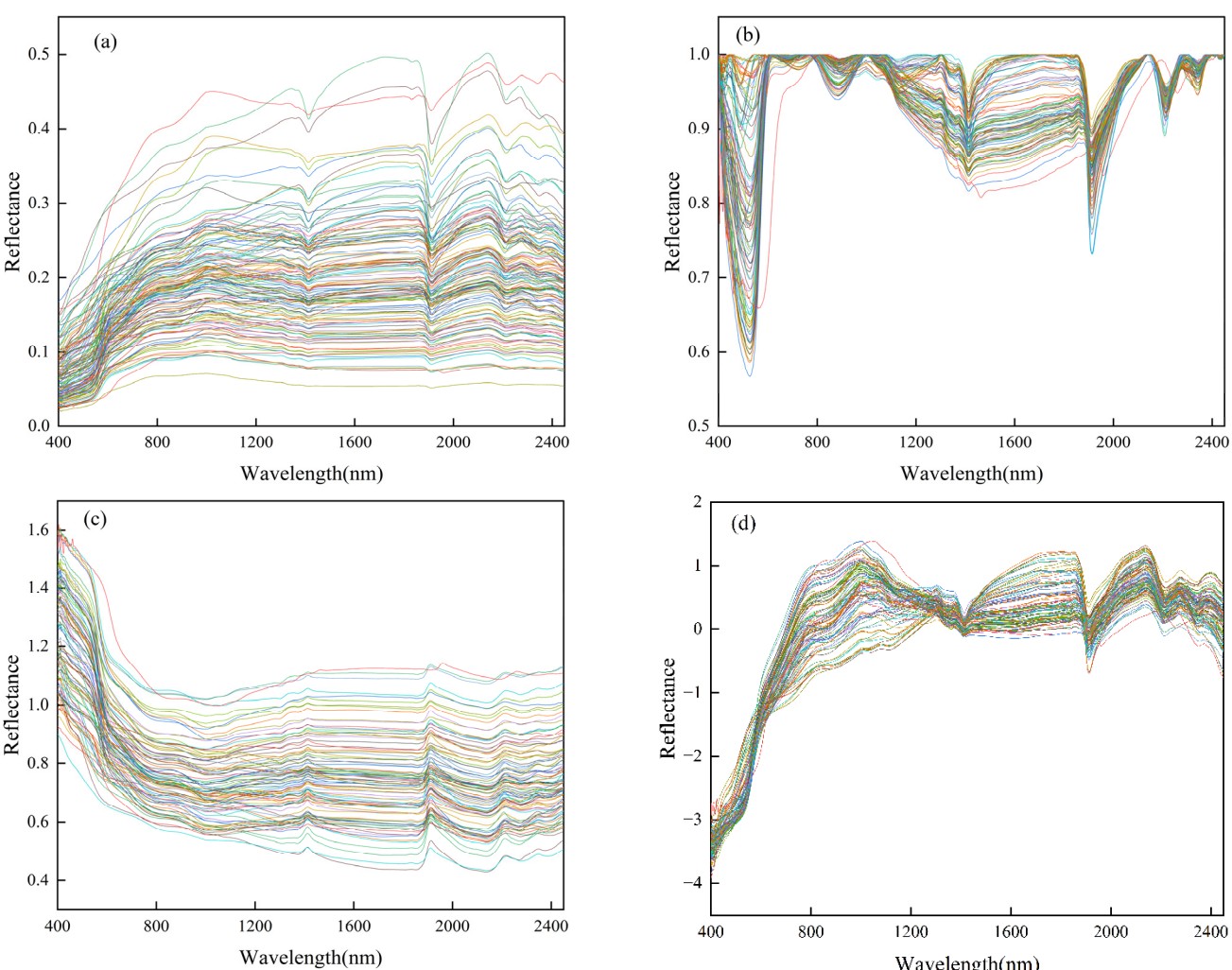

**Figure 3.** Transformed spectra of the soil samples: (**a**) original spectra. (**b**) continuum removal. (**c**) absorbance. (**d**) SNV transformed reflectance.

### 3.2.3. Fractional Order Differential

The FOD spectra of the calibration dataset are shown in Figure 5. The absorption features at 1400, 1900 and 2200 nm were more obvious, but the absorption bands were wider and overlapped. When the order was gradually increased from 0 to 1, the differential curve of each order slowly approximated the differential curve of the 1-order, and the sensitivity of the differential result to the slope of the reflectivity curve increased [49]. The three absorption features of the water molecule vibration at 1400, 1900 and 2200 nm became increasingly apparent; at the same time, there were two positive peaks at 420 and 570 nm and one negative peak at 470 nm and the absorption band at 1400 nm changed from one negative peak to one positive and one negative peak. As the order increased, the spectral reflectance values gradually approached 0, which indicates that the baseline drift and overlapping peaks were eliminated [19]. At the same time, the absorption valleys at 1900 and 2200 nm gradually changed to a positive and a negative peak, respectively. Compared to the original spectra, the FOD spectra can show changes in spectral detail and improve the resolution of the spectral curve.

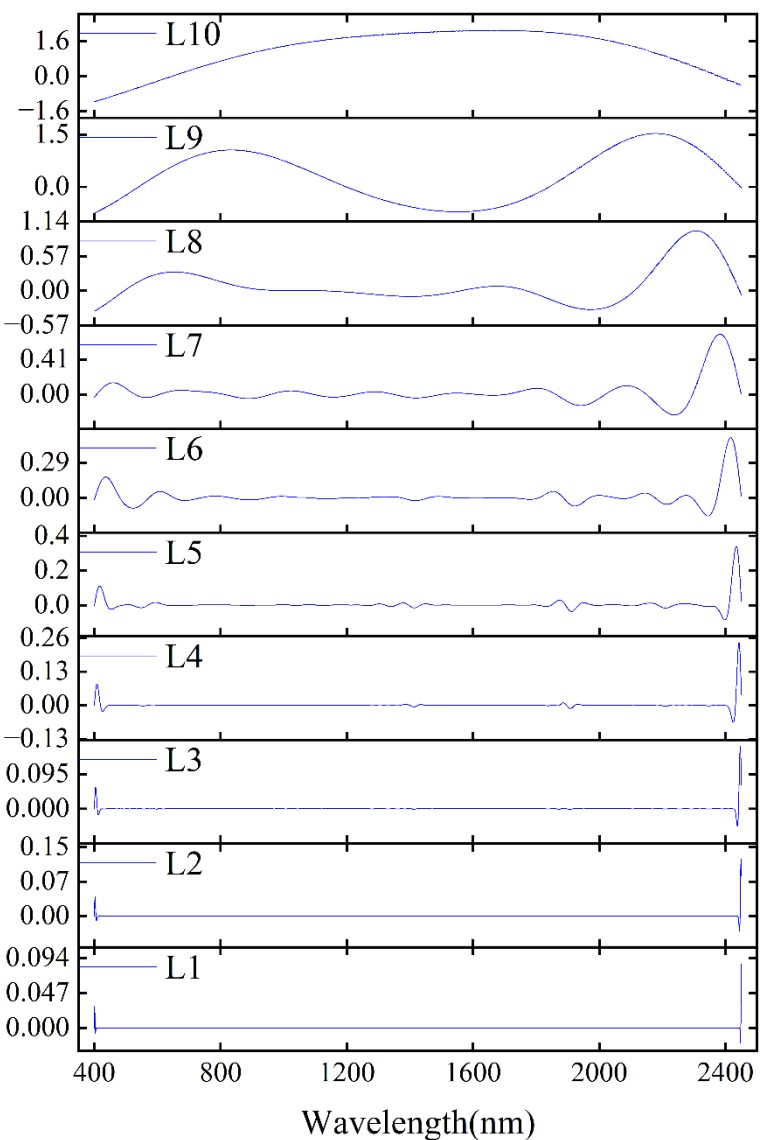

**Figure 4.** CWT spectra at different scales.

*3.3. Correlation of Transformation Spectra with Iron Oxides*

In order to observe the effect of different spectral transformations on the original spectrum, the correlation analysis of three transformed spectra with iron oxide was performed.

### 3.3.1. Correlation of Conventional Transformations with Iron Oxide

The conventional transformation spectra were correlated with the iron oxide content on a wavelength-by-wavelength basis. The results are shown in Figure 6. The bands that passed the 0.01 significance test for CR were mainly around 400–600, 1200–1900, 2100, 2300 and 2400 nm. The overall correlation coefficient curve was similar to that of CR, while log (1/R) passed the 0.01 significance test for all bands. Table 2 shows that log (1/R) achieved the highest correlation coefficient of 0.606 compared to SNV and CR, followed by SNV with a maximum correlation coefficient of −0.590 and 1622 wavelengths passing the 0.01 significance test. The lowest correlation coefficient was obtained for CR with a value of 0.573 and 1255 wavelengths passing the 0.01 significance test.

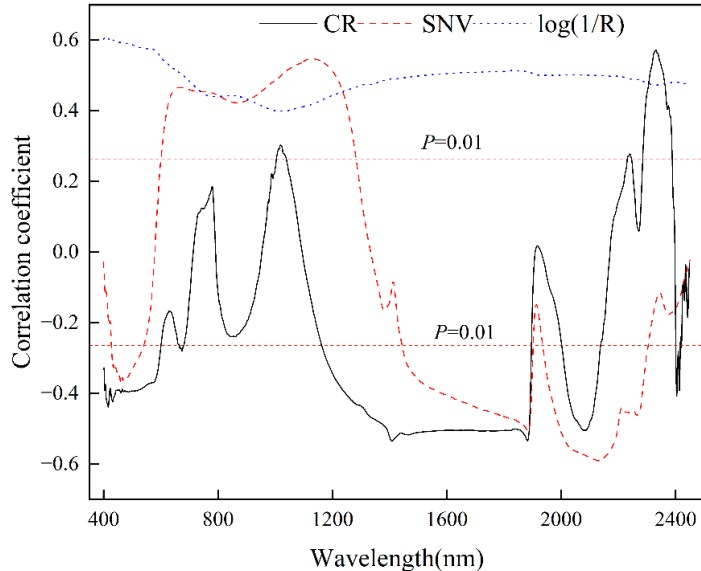

**Figure 5.** The FOD spectra of soil samples in the calibration dataset: (**a**) original spectra; (**b**) 0.25-order, (**c**) 0.5-order; (**d**) 0.75-order; (**e**) 1-order; (**f**) 1.25-order; (**g**) 1.5-order; (**h**) 1.75-order; (**i**) 2-order.

**Figure 6.** Correlation coefficient between spectra and iron oxide content.

**Table 2.** The number of wavelengths passing the significance test ($p < 0.01$) and the maximum correlation coefficient for different spectra.

| Spectral Transformation Name | Number of Significant Wavelengths | Maximum Correlation Coefficient |
|---|---|---|
| CR | 1255 | 0.573 |
| Log (1/R) | 2051 | **0.606** |
| SNV | 1622 | −0.590 |

### 3.3.2. Correlation of Continuous Wavelet Transform with Iron Oxide

The wavelet coefficients at each scale were correlated with the iron oxide content of the soil. The heat map of the coefficient of determination of the wavelet coefficient and the iron oxide content is shown in Figure 7. The higher determination coefficients of wavelet coefficients and iron oxide content were mainly distributed in the visible band at the L3, L4, L5 and L6, with the highest determination coefficient reaching 0.367, indicating that the effective information was mainly concentrated at the L3, L4, L5 and L6. At the L1 and L2, the determination coefficients were lower, indicating that some spectral features disappeared and the effective information was less. The number of wavelengths passing the 0.01 significance test and the maximum correlation coefficient for each scale are shown in Table 3. The maximum correlation coefficient of CWT was 0.606 (L6). Meanwhile, the absolute value of the correlation coefficient between wavelet coefficient and iron oxide content of each scale showed a trend of increasing and then decreasing, and the number of its significant wavelengths showed a gradual increase.

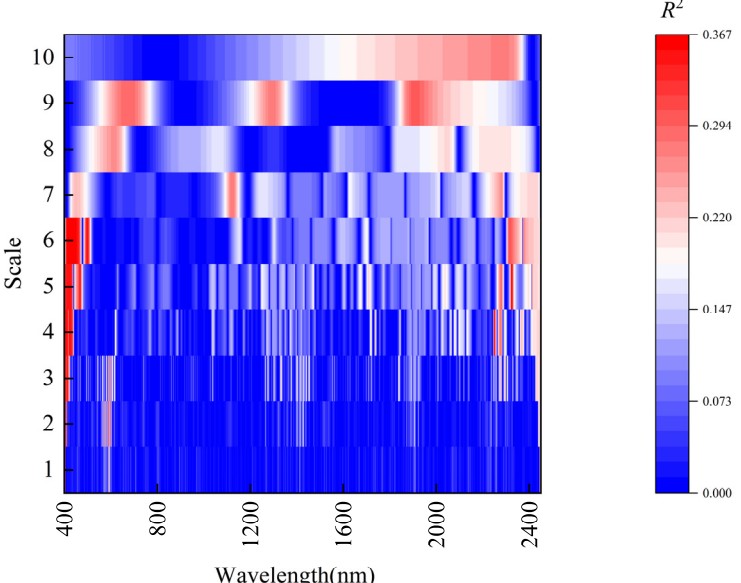

**Figure 7.** The heat map of the coefficient of determination of the wavelet coefficient and the iron oxide content.

### 3.3.3. Correlation of Fractional Order Differential with Iron Oxide

Figure 8 shows the correlation coefficients between the different fractional order spectra and the soil iron oxide content in the calibration dataset samples. All bands in the OS (Figure 8a) were negatively correlated with soil iron oxide content. The full range of bands passed the 0.01 significance test. The correlation coefficients between the original spectra and the soil iron oxide content varied smoothly with the wavelength. As the order increased, many positive and negative correlation peaks gradually appeared, and positive and negative correlations occurred for adjacent wavelengths. As can be seen in Table 4, the wavelengths that passed the 0.01 significance test gradually decreased as the order increased, with the maximum correlation reached its maximum (−0.620) at order

0.75 (Figure 8d), while the maximum absolute correlation for the original reflectance was only equal to 0.589. The 1-order and 2-order differentials showed lower correlations than the other order differentials (Figure 8e,i). The FOD provides additional detailed spectral variation information compared to the original spectra (0-order) and the commonly used integer order (1-order and 2-order).

**Table 3.** The number of wavelengths passing the significance test (*p* < 0.01) and the maximum correlation coefficient for different scale in CWT.

| Spectral Transformation Name | Wavelet Decomposition Scale | Number of Significant Wavelengths | Maximum Correlation Coefficient |
|---|---|---|---|
| | L1 | 92 | −0.590 |
| | L2 | 171 | −0.593 |
| | L3 | 395 | **0.606** |
| | L4 | 663 | −0.602 |
| CWT | L5 | 1136 | 0.603 |
| | L6 | 1056 | −0.604 |
| | L7 | 1320 | 0.527 |
| | L8 | 1357 | −0.511 |
| | L9 | 1273 | −0.548 |
| | L10 | 1447 | −0.523 |

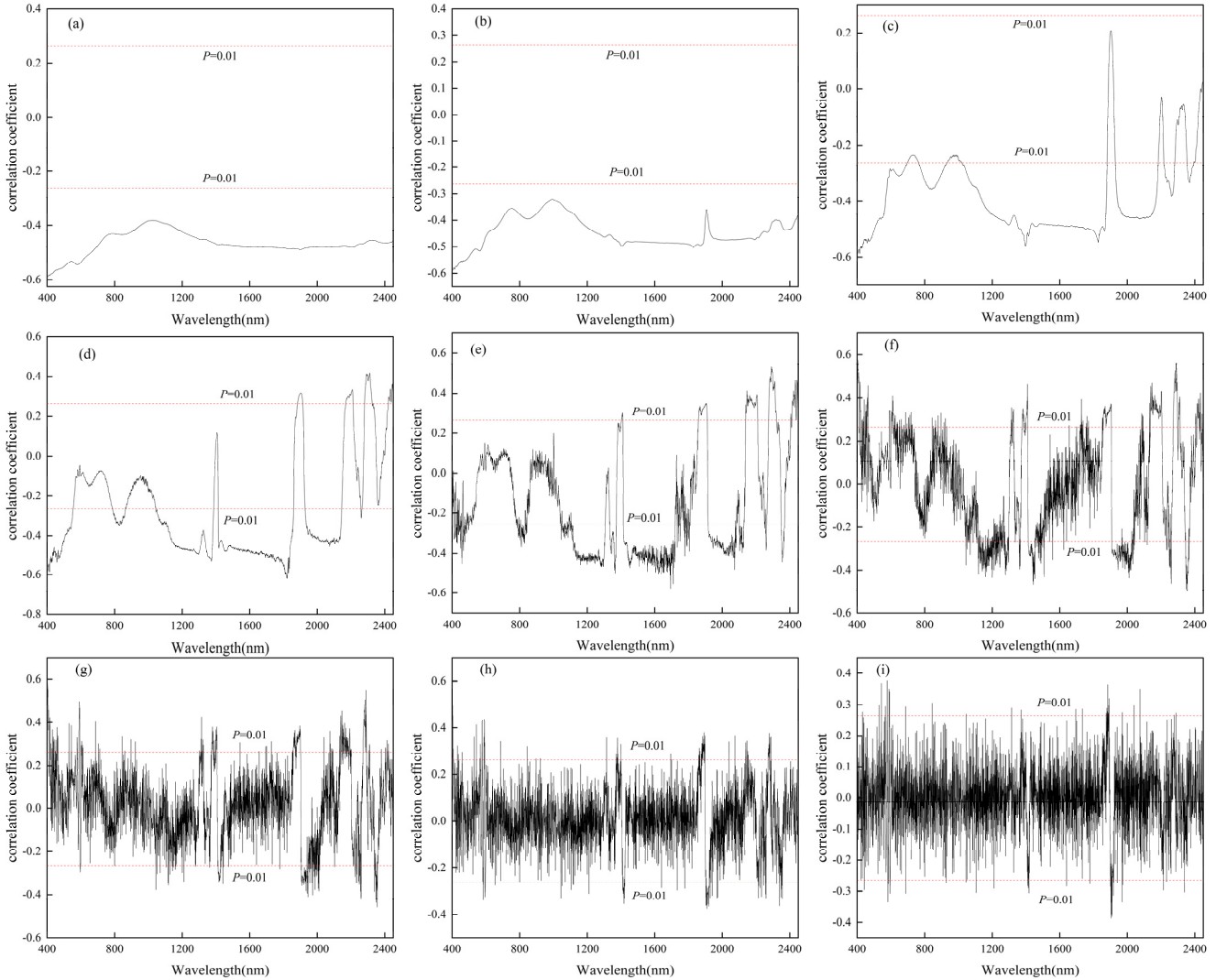

**Figure 8.** The correlation coefficients between the FOD spectra in the calibration dataset and the soil iron oxide content: (**a**) original spectra; (**b**) 0.25-order, (**c**) 0.5-order; (**d**) 0.75-order; (**e**) 1-order; (**f**) 1.25-order; (**g**) 1.5-order; (**h**) 1.75-order; (**i**) 2-order.

**Table 4.** The number of wavelengths passing the significance test ($p < 0.01$) and the maximum correlation coefficient for different order in FOD.

| Spectral Transformation Name | Number of Significant Wavelengths | Maximum Correlation Coefficient |
| --- | --- | --- |
| 0-order | 2051 | −0.589 |
| 0.25-order | 2051 | −0.589 |
| 0.5-order | 1683 | −0.590 |
| 0.75-order | 1394 | **−0.620** |
| 1-order | 1138 | −0.578 |
| 1.25-order | 665 | 0.592 |
| 1.5-order | 341 | 0.593 |
| 1.75-order | 132 | 0.594 |
| 2-order | 57 | −0.387 |

*3.4. Characteristic Wavelength Selection*

If the full wavelength band is used directly as an input variable for modelling, not only is it too inefficient but it may also reduce the accuracy of the model. In this study, CARS was used for the selection of the characteristic wavelengths. As the Monte Carlo sampling method is unstable, the results varied over multiple runs. Therefore, in this study, CARS was cycled through 50 experiments, and the wavelengths with frequencies up to 20 or 30 times in the results obtained were used as the characteristic wavelengths, and their frequency domain thresholds were selected according to the actual situation. The results of the characteristic wavelengths selected according to the CARS algorithm are shown in Figure 9. It was found that most of the wavelengths selected using CARS under the 0.5-order differential transform were distributed around 400 nm, 440 nm and 900 nm, which is consistent with the absorption peak of iron, and the other bands were distributed at 1900 nm and 2200 nm, which was due to the influence of various functional groups. Too few wavelengths were screened at 0-order and 0.25-order, which may have led to later modelling effects being reduced. Wavelengths greater than the 1-order differential screening (1.5-order, 1.75-order and 2-order) were distributed over almost the whole waveband, especially at 600–800 nm, which is considered by previous authors to be the characteristic waveband of organic matter [50]. The L1, L2, L3, CR and log (1/R) were also distributed in the characteristic band of organic matter.

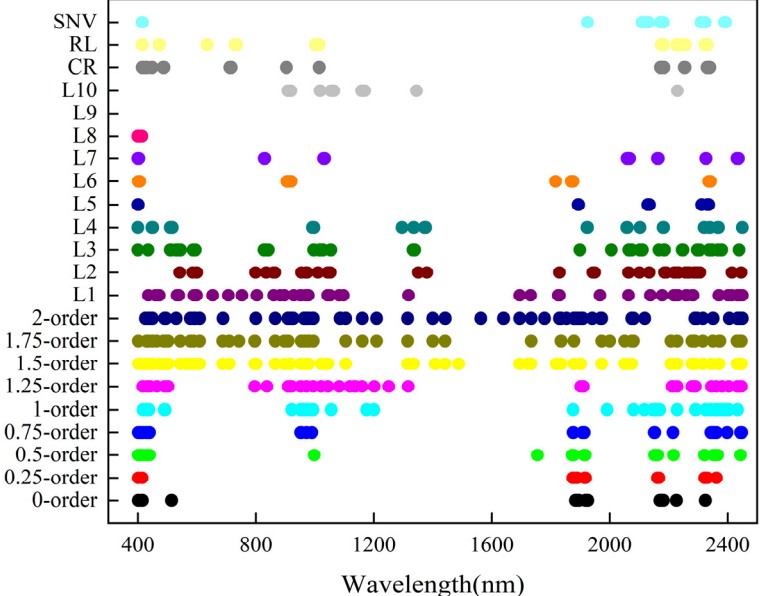

**Figure 9.** Diagram of the selection of characteristic wavelengths for different spectral transformations.

### 3.5. Model Construction and Evaluation Using the Full Spectrum

The BPNN and SVR models for estimating soil iron oxide content were constructed based on the full spectrum. As shown in Tables 5 and 6, of the conventional transformations, CR obtained the best prediction accuracy using BPNN with an RPIQ of 2.480, followed by log (1/R) and SNV with RPIQ values of 2.152 and 2.277, respectively. All three models can be considered good models. CR obtained the best prediction accuracy using SVR with an RPIQ of 2.092, followed by log (1/R) and SNV with RPIQ values of 1.813 and 1.898, respectively. In CWT, L7 obtained a good prediction accuracy using BPNN with an RPIQ value of 2.628, which can be considered as an excellent model. However, CWT used SVR to construct the model and achieved the highest accuracy at L4 with an RPIQ value of 2.440. In the FOD, a superior accuracy was obtained for 0.75-order using BPNN and SVR with RPIQ values of 3.045 and 2.529, respectively. By comparing three spectral transforms and two model construction methods, FOD and BPNN showed better performances.

**Table 5.** The results of the BPNN estimation of soil iron oxide content using the full spectrum.

| Spectral Transformation Name | Calibration Dataset | | Validation Dataset | | |
| :---: | :---: | :---: | :---: | :---: | :---: |
| | $R^2$ | RMSE/(g·kg$^{-1}$) | $R^2$ | RMSE/(g·kg$^{-1}$) | RPIQ |
| CR | **0.795** | **5.096** | **0.671** | **8.168** | **2.480** |
| Log (1/R) | 0.753 | 6.624 | 0.624 | 9.416 | 2.152 |
| SNV | 0.876 | 3.378 | 0.579 | 8.897 | 2.277 |
| L1 | 0.621 | 7.367 | 0.161 | 12.399 | 1.634 |
| L2 | 0.653 | 6.792 | 0.199 | 14.474 | 1.400 |
| L3 | 0.705 | 6.419 | 0.305 | 15.558 | 1.302 |
| L4 | 0.744 | 5.580 | 0.539 | 9.992 | 2.028 |
| L5 | 0.819 | 4.602 | 0.463 | 10.172 | 1.992 |
| L6 | 0.897 | 3.446 | 0.580 | 9.124 | 2.221 |
| L7 | **0.728** | **5.734** | **0.707** | **7.711** | **2.628** |
| L8 | 0.323 | 10.385 | 0.199 | 13.722 | 1.476 |
| L9 | 0.788 | 4.981 | 0.276 | 12.529 | 1.617 |
| L10 | 0.734 | 5.517 | 0.501 | 9.687 | 2.092 |
| 0-order | 0.836 | 4.358 | 0.488 | 9.681 | 2.093 |
| 0.25-order | 0.812 | 4.667 | 0.670 | 7.887 | 2.569 |
| 0.5-order | 0.887 | 3.621 | 0.727 | 7.622 | 2.658 |
| 0.75-order | **0.876** | **3.763** | **0.782** | **6.654** | **3.045** |
| 1-order | 0.902 | 3.347 | 0.641 | 8.527 | 2.376 |
| 1.25-order | 0.750 | 5.458 | 0.454 | 10.920 | 1.855 |
| 1.5-order | 0.661 | 6.795 | 0.384 | 11.106 | 1.824 |
| 1.75-order | 0.634 | 6.741 | 0.203 | 14.039 | 1.443 |
| 2-order | 0.663 | 6.414 | 0.155 | 13.498 | 1.501 |

### 3.6. Model Construction and Evaluation Using Characteristic Wavelengths

The BPNN and SVR models for estimating soil iron oxide content were constructed based on different spectra. As shown in Tables 7 and 8, the model constructed using SNV was better than OS in BPNN, with an RPIQ of 3.151. This is related to the fact that SNV can reduce the non-specific scattering noise on the sample surface. The model performance of L4, L6 and L7 was better than that of the original spectra with RPIQ values of 3.035, 3.199 and 3.023, respectively. The performance of the L1, L2, L8 and L9 models was not as good as their RPIQ values were less than 2 and the predictive power of the models was poor. This indicates that a scale that is either too low or too high affects the accuracy of the model, with too low a scale resulting in large noise and too high a scale smoothing out some important absorption features. The prediction accuracy of the FOD spectra model varied greatly for different orders. The maximum value of the validation dataset RPIQ was 3.686 and the minimum value was only 1.729. This result shows that it is necessary to choose the right order of the FOD spectra for modelling. The model constructed using CR was better than the log (1/R), SNV and OS in the SVR with a validation set RPIQ of 2.526.

In the CWT, it was the L4 that performed better with an RPIQ of 2.748, while the FOD was better for the model constructed at the 0.75-order with an RPIQ of 2.647. Comparing all the transform spectra, it was found that the BPNN model constructed using the 0.5-order FOD spectra performed significantly better, with the highest $R^2$ and RPIQ and the lowest RMSE in the validation dataset. This result suggests that the FOD combined with BPNN has more potential for estimating soil iron oxide content.

**Table 6.** The results of the SVR estimation of soil iron oxide content using the full spectrum.

| Spectral Transformation Name | Calibration Dataset | | Validation Dataset | | |
| --- | --- | --- | --- | --- | --- |
| | $R^2$ | RMSE/(g·kg$^{-1}$) | $R^2$ | RMSE/(g·kg$^{-1}$) | RPIQ |
| CR | **0.591** | **6.805** | **0.480** | **9.687** | **2.092** |
| Log (1/R) | 0.579 | 6.932 | 0.308 | 11.174 | 1.813 |
| SNV | 0.522 | 7.386 | 0.368 | 10.677 | 1.898 |
| L1 | 0.548 | 7.175 | 0.147 | 12.404 | 1.633 |
| L2 | 0.752 | 5.322 | 0.332 | 10.982 | 1.845 |
| L3 | 0.780 | 5.005 | 0.537 | 9.142 | 2.216 |
| L4 | **0.744** | **5.396** | **0.618** | **8.305** | **2.440** |
| L5 | 0.699 | 5.857 | 0.615 | 8.339 | 2.430 |
| L6 | 0.651 | 6.309 | 0.591 | 8.594 | 2.358 |
| L7 | 0.619 | 6.591 | 0.521 | 9.300 | 2.178 |
| L8 | 0.597 | 6.776 | 0.487 | 9.616 | 2.107 |
| L9 | 0.604 | 6.720 | 0.490 | 9.590 | 2.113 |
| L10 | 0.403 | 8.247 | 0.264 | 11.522 | 1.758 |
| 0-order | 0.594 | 6.805 | 0.346 | 10.860 | 1.866 |
| 0.25-order | 0.694 | 5.911 | 0.532 | 9.193 | 2.204 |
| 0.5-order | 0.808 | 4.681 | 0.621 | 8.274 | 2.449 |
| 0.75-order | **0.769** | **5.129** | **0.644** | **8.010** | **2.529** |
| 1-order | 0.733 | 5.519 | 0.572 | 8.791 | 2.304 |
| 1.25-order | 0.803 | 4.743 | 0.573 | 8.781 | 2.307 |
| 1.5-order | 0.814 | 4.602 | 0.348 | 10.845 | 1.868 |
| 1.75-order | 0.544 | 7.207 | 0.149 | 12.391 | 1.635 |
| 2-order | 0.666 | 6.170 | 0.105 | 12.706 | 1.595 |

**Table 7.** The results of the BPNN estimation of soil iron oxide content using characteristic wavelengths.

| Spectral Transformation Name | Number of Characteristic Wavelengths | Calibration Dataset | | Validation Dataset | | |
| --- | --- | --- | --- | --- | --- | --- |
| | | $R^2$ | RMSE/(g·kg$^{-1}$) | $R^2$ | RMSE/(g·kg$^{-1}$) | RPIQ |
| CR | 47 | 0.798 | 4.850 | 0.732 | 6.955 | 2.913 |
| Log (1/R) | 43 | 0.824 | 4.640 | 0.695 | 7.422 | 2.730 |
| SNV | 31 | **0.795** | **4.873** | **0.772** | **6.431** | **3.151** |
| L1 | 62 | 0.623 | 7.225 | 0.283 | 12.347 | 1.641 |
| L2 | 55 | 0.878 | 3.752 | 0.499 | 9.539 | 2.124 |
| L3 | 68 | 0.913 | 3.247 | 0.607 | 8.903 | 2.276 |
| L4 | 48 | 0.779 | 5.209 | 0.763 | 6.676 | 3.035 |
| L5 | 20 | 0.676 | 6.346 | 0.668 | 8.019 | 2.527 |
| L6 | 35 | **0.788** | **5.239** | **0.787** | **6.334** | **3.199** |
| L7 | 26 | 0.802 | 4.760 | 0.753 | 6.703 | 3.023 |
| L8 | 6 | 0.396 | 8.347 | 0.382 | 10.573 | 1.916 |
| L9 | 6 | 0.568 | 7.103 | 0.448 | 11.618 | 1.744 |
| L10 | 10 | 0.541 | 7.301 | 0.518 | 9.347 | 2.168 |
| 0-order | 44 | 0.731 | 5.538 | 0.737 | 7.082 | 2.847 |
| 0.25-order | 20 | 0.724 | 5.655 | 0.719 | 7.703 | 2.630 |
| 0.5-order | 38 | **0.903** | **3.376** | **0.851** | **5.497** | **3.686** |
| 0.75-order | 32 | 0.836 | 4.424 | 0.832 | 6.162 | 3.288 |
| 1-order | 41 | 0.917 | 4.122 | 0.717 | 7.713 | 2.627 |
| 1.25-order | 47 | 0.943 | 2.569 | 0.702 | 7.817 | 2.592 |
| 1.5-order | 77 | 0.891 | 3.574 | 0.544 | 9.696 | 2.090 |
| 1.75-order | 55 | 0.764 | 6.086 | 0.431 | 10.401 | 1.948 |
| 2-order | 68 | 0.905 | 3.340 | 0.246 | 11.721 | 1.729 |

**Table 8.** The results of the SVR estimation of soil iron oxide content using characteristic wavelengths.

| Spectral Transformation Name | Number of Characteristic Wavelengths | Calibration Dataset | | Validation Dataset | | |
|---|---|---|---|---|---|---|
| | | R² | RMSE/(g·kg⁻¹) | R² | RMSE/(g·kg⁻¹) | RPIQ |
| CR | 47 | **0.701** | **5.838** | **0.644** | **8.022** | **2.526** |
| Log (1/R) | 43 | 0.421 | 8.128 | 0.363 | 10.729 | 1.889 |
| SNV | 31 | 0.712 | 5.728 | 0.494 | 9.559 | 2.120 |
| L1 | 62 | 0.628 | 6.511 | 0.203 | 11.993 | 1.689 |
| L2 | 55 | 0.866 | 3.910 | 0.432 | 10.126 | 2.001 |
| L3 | 68 | 0.834 | 4.351 | 0.517 | 9.341 | 2.169 |
| L4 | 48 | **0.716** | **4.549** | **0.652** | **7.924** | **2.748** |
| L5 | 20 | 0.678 | 6.062 | 0.599 | 8.512 | 2.380 |
| L6 | 35 | 0.716 | 5.688 | 0.606 | 8.428 | 2.404 |
| L7 | 26 | 0.695 | 5.897 | 0.602 | 8.471 | 2.392 |
| L8 | 6 | 0.311 | 8.864 | 0.272 | 11.462 | 1.768 |
| L9 | 6 | 0.232 | 9.375 | 0.225 | 11.828 | 1.713 |
| L10 | 10 | 0.376 | 8.434 | 0.292 | 11.306 | 1.792 |
| 0-order | 44 | 0.512 | 7.457 | 0.415 | 10.274 | 1.972 |
| 0.25-order | 20 | 0.689 | 5.946 | 0.563 | 8.878 | 2.282 |
| 0.5-order | 38 | 0.766 | 5.165 | 0.651 | 7.934 | 2.554 |
| 0.75-order | 32 | **0.727** | **5.576** | **0.675** | **7.655** | **2.647** |
| 1-order | 41 | 0.686 | 5.985 | 0.622 | 8.255 | 2.455 |
| 1.25-order | 47 | 0.878 | 3.725 | 0.634 | 8.126 | 2.493 |
| 1.5-order | 77 | 0.826 | 4.455 | 0.425 | 10.182 | 1.990 |
| 1.75-order | 55 | 0.768 | 5.147 | 0.214 | 10.401 | 1.948 |
| 2-order | 68 | 0.607 | 6.693 | 0.308 | 10.273 | 1.972 |

## 4. Discussion

Soil spectral information is a comprehensive reflection of the soil, which is mainly influenced by soil organic matter, iron oxide, soil texture and pH. The spectral features of soil organic matter and iron oxide in the visible NIR band often overlap [51], resulting in the absorption features of iron oxide in the OS being easily obscured by organic matter [14]. As a result, the inversion of iron oxide using OS may not achieve the expected accuracy. Transformation of spectra is an important tool to improve the predictive power of models [52], and different spectral transformations have different effects in enhancing correlation and highlighting spectral features. In this study, we used three types of spectral transform methods: the conventional spectral transform, CWT and FOD. Among the conventional spectral transforms, the SNV obtained a better prediction accuracy, which may be related to the fact that the SNV transform eliminates the effects of soil grain size, soil surface scattering and light range transformation on reflectance [53]. However, Tan Jie et al. [54] predicted iron oxide in mountainous red soils and found that the CR transform had the highest prediction accuracy compared to the differential transform. The CWT can perform multi-scale decomposition in the time and frequency domains [55,56], and invert the physicochemical properties of soils by finding wavelet coefficients at different scales [57]. The decomposition of the OS using the CWT reveals that the high-frequency information of the wavelet decomposition reflects the main absorption characteristics of the soil hyperspectral, and the sensitivity of the high-frequency information increases with the degree of wavelet decomposition [58]. In this study, the CWT obtained the best prediction results at the L6. However this differs from previous studies that have analyzed the copper content of chicory leaves and found that their spectra were CWT transformed to have optimum scales of L3, L4 and L5 [59]. Differential transformations can reduce the noise and enhance the spectral features of a spectrum [60]. However, traditional integer order differentials cannot capture detailed spectral information [17]. The FOD is gradually being applied to the study of soil spectra with good results [17,61]. The FOD can vary the spectral reflectance at small intervals with different degrees of curvature, thus capturing spectral features that cannot be captured by integer order differential [62]. In this study, the prediction accuracy reached its maximum at 0.5-order as the order increases. Previous studies have also found similar results when using FOD to estimate SOM and moisture content [63]. The reason may be that FOD offers a better balance between spectral resolution, spectral information and

noise than integer order spectra [64]. When the order is greater than 1, the amount of noise exceeds the amount of spectral information, which has a negative impact on the accuracy of the model [65]. Overall, all three types of transformations were effective in improving the prediction accuracy of the model, but the FOD had the best prediction accuracy, and the lower order was more advantageous than the higher order.

Due to the wavelength redundancy of hyperspectral, characteristic wavelength extraction is necessary. Different researchers have differed in their methods of characteristic wavelength selection, including the selection of characteristic wavelengths by correlation analysis using different spectral transformations with iron oxide content [22,29], or by stepwise regression and principal component analysis based on correlation analysis [66]. This study used CARS for characteristic wavelength selection. Due to the instability of CARS, the CARS algorithm was run 50 times in a loop by us to select wavelengths with a frequency of 20 or 30 times as feature wavelengths. Too many or too few characteristic wavelengths can affect the prediction accuracy of the model. From Figure 9, we can see that the number of wavelengths screened out by 0.5 order differential was 38, most of which were distributed around 400 nm, 440 nm and 900 nm, which is consistent with the absorption peak of iron, and the other wavelengths were distributed at 1900 nm and 2200 nm, which was due to the influence of various functional groups, consistent with previous studies. The 1.5-order, 1.75-order, 2-order, L1, L2 and L3 screened out too many wavelengths and the selected wavelengths were distributed in the organic matter characteristic band from 600 to 800 nm. L8, L9 and L10, on the other hand, screened out too few wavelengths and filtered out many wavelengths that were beneficial to the model, resulting in lower prediction accuracy of the model. Therefore, this method can be used as an effective wavelength screening method.

Previous studies have typically used linear models to predict the iron oxide content of soils [23,67], and this study utilized the more widely used BPNN and SVR to construct the models. Neural networks have good approximation properties and generalization capabilities, but often require a large amount of sample data to build excellent models. However, when such networks are applied to small sample data, the input to the model needs to be pre-processed to achieve good prediction accuracy. In this paper, with a small number of samples ($n$ = 135), a series of pre-processing such as the above-mentioned data set partitioning, spectral transformation and extraction of characteristic wavelengths were performed. A BPNN with only one hidden layer was also constructed, which is a simpler network structure and belongs to a shallow neural network. This can avoid the overfitting of the model or a poor generalization performance when the sample data are small. Han lei et al. [68] used BPNN to analyze small sample data ($n$ = 90) and found that BPNN had a higher prediction accuracy compared to PLSR, but also found that there was a corresponding increase in accuracy as the sample size increased. In this study, the results are shown in Tables 7 and 8. The BPNN achieved the best prediction results for the 0.5-order differential transformation and the use of characteristic wavelengths ($R^2$ = 0.851, RMSE = 5.497 and RPIQ = 3.686). This differs from the previous study [66], which concluded that the first-order differential can effectively improve the prediction accuracy of the model. The SVR achieved the best results that were obtained at 0.75-order ($R^2$ = 0.675, RMSE = 7.655 and RPIQ = 2.647). Comparing the two methods of constructing the model, BPNN achieved the best model prediction capability at 0.5-order.

## 5. Conclusions

In this paper, indoor hyperspectral data of surface soils from the southern edge of the Dinosaur Valley in Lufeng, Yunnan, were combined with laboratory data of iron oxide content to perform an inversion of iron oxide content in the region. To verify the predictive power of FOD for iron oxide, the conventional spectral transform and CWT were used for comparison. It was found that the maximum correlation of FOD was stronger than that of the conventional spectral transformation and CWT. The accuracy of the model constructed by full spectrum and characteristic wavelengths was also compared, and it was found that

the accuracy of full spectrum was lower than that using characteristic wavelengths, which indicates that it is necessary to carry out the selection of characteristic wavelengths before the model construction. The FOD achieved the best results among the different modelling methods, with the 0.5-order-BPNN having the strongest predictive power. It indicates that the FOD can obtain more detailed spectral features and effectively improve the prediction ability in soil iron oxide.

The current work was all carried out indoors, and although a high accuracy was obtained, it was limited to small scales. In the future, however, there will be a trend towards using hyperspectral satellites to explore soil spectra at large scales in estimating iron oxide content and describing its spatial distribution.

**Author Contributions:** Conceptualization, H.Z. and S.G.; methodology, H.Z.; software, H.Z.; validation, H.Z., L.H. and J.W.; investigation, H.Z.; resources, S.G. and X.Y.; data curation, H.Z., L.H., J.W. and S.L.; writing—original draft preparation, H.Z.; writing—review and editing, S.G.; funding acquisition, S.G. All authors have read and agreed to the published version of the manuscript.

**Funding:** This research was funded by the National Natural Science Foundation of China, grant number 41861054.

**Institutional Review Board Statement:** Not applicable.

**Informed Consent Statement:** Not applicable.

**Data Availability Statement:** Not applicable.

**Acknowledgments:** We are very grateful to Rui Bi for his assistance in writing the thesis and to Xingping Wen for his experimental assistance.

**Conflicts of Interest:** The authors declare no conflict of interest.

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
