# Peer review of "Application of a Fractional Order Differential to the Hyperspectral Inversion of Soil Iron Oxide"

_agriculture, doi:10.3390/agriculture12081163_

Round 1

Reviewer 1 Report

Please find my comments and suggestions in the PDF attached file.

Kind regards

Reviewer 2 Report

In this manuscript, soil iron oxide is estimated using hyperspectral measurements. Topic is interesting, but authors must consider the following comments to improve the quality of the manuscript:

·         Lines 16 to 21: it is a large sentence. Please revise it.

·         Line60: Please revise carefully spacing errors.

·         Data processing section: I would like to see a methodology section with a workflow.

·         Please define variables of equations.

·         Line 181: did you use random method to select samples as training and test ones?

·         Line 192: X and Y are used in different equations, but their definition in each one is different. It is recommended to use other symbols instead of X and Y.

·         Conclusion: Please insert some future works in this section.

·         Is it possible to use satellite remote sensing images and your field measurements to estimate the soil iron oxide?

Round 2

Reviewer 1 Report

I have no further comment, other than that the responses to the comments are relevant and the revised version of the manuscript has been significantly improved.

Reviewer 2 Report

Accept as it is.